# Data splitting to avoid information leakage with DataSAIL

Roman Joeres [1,2,3,4] ✉, David B. Blumenthal [5,7] & Olga V. Kalinina [1,2,6,7]

Information leakage is an increasingly important topic in machine learning research for biomedical applications. When information leakage happens during a model's training, it risks memorizing the training data instead of learning generalizable properties. This can lead to inflated performance metrics that do not reflect the actual performance at inference time. We present DataSAIL, a versatile Python package to facilitate leakage-reduced data splitting to enable realistic evaluation of machine learning models for biological data that are intended to be applied in out-of-distribution scenarios. DataSAIL is based on formulating the problem to find leakage-reduced data splits as a combinatorial optimization problem. We prove that this problem is NP-hard and provide a scalable heuristic based on clustering and integer linear programming. Finally, we empirically demonstrate DataSAIL's impact on evaluating biomedical machine learning models.

Supervised machine learning (ML) is one of the fastest-growing research fields, leading to advances in many computer and life science domains. Many bioinformatics fields benefit from using ML models, e.g., molecular property prediction[1] and drug-target interaction prediction[2].

For successful deployment of these ML models in real-world use cases, it is crucial that the reported performance estimates reliably represent model performance during inference. If the test set used for model evaluation does not represent the data used at inference time, the model can show inflated performance scores during testing, impeding successful model deployment in a real-world use case. Such misrepresentation can happen when the model uses information from the training set at test time, although this information is not available during inference. This phenomenon is called information leakage or data leakage[3,4]. Recent studies show that information leakage is a highly relevant problem in many subfields of ML-based research, leading to inflated performances and overoptimistic conclusions in biomedical ML research[4–6] and beyond[7].

The simplest form of information leakage is having the same samples in multiple folds of the data split. This is easy to control, and

most common splitting techniques avoid this by removing duplicate data points. Another type of information leakage that is more complex to detect can occur when similarities between data points in the training and in the test sets are larger than similarities between data points in the training set and in the data that one intends to use during inference[4]. In such a case, an ML model is benchmarked on test data that is in-distribution with respect to the training data, although its intended use case is to yield reliable predictions also for out-of-distribution (OOD) data. Hence, a model may perform well on the test data by relying on similarity-based shortcuts that do not generalize to the intended real-world application scenario.

Especially for biomolecular data that exhibit complex dependency structures, one can easily fall into this trap by following a standard strategy in the ML community to randomly split a benchmarking dataset into training, validation, and test folds. For instance, it has been shown that this problem pervades the field of research on deep learning models to predict protein-protein interaction (PPI) from protein sequences[8–10]. While many of these models perform excellently when evaluated on the random data splits used in the original publications, performance often becomes close to random when

[1]Helmholtz Institute for Pharmaceutical Research Saarland (HIPS), Helmholtz Centre for Infection Research (HZI), Saarbrücken, Germany. [2]Center for Bioinformatics, Saarland University, Saarbrücken, Germany. [3]Department of Chemistry and Molecular Biology, University of Gothenburg, Gothenburg, Sweden. [4]Wallenberg Centre for Molecular and Translational Medicine, University of Gothenburg, Gothenburg, Sweden. [5]Department Artificial Intelligence in Biomedical Engineering, Friedrich-Alexander-Universität Erlangen-Nürnberg, Erlangen, Germany. [6]Medical Faculty, Saarland University, Homburg, Germany. [7]These authors jointly supervised this work: David B. Blumenthal, Olga V. Kalinina. ✉e-mail: roman.joeres@helmholtz-hips.de

evaluated on protein pairs with low homology to the training data that represents the desired use case to predict PPIs for poorly characterized proteins[10]. Another biomedical ML problem for which similar pitfalls have been described is the problem of predicting the deleteriousness of missense variants[11,12]. For this problem, it has been shown that when variants that are similar in that they affect the same protein are assigned to different splits, ML models can achieve excellent test performances by relying on protein-level shortcuts (e.g., a simple protein-level majority vote based on the variants in the training fold). Such models then generalize poorly to variants of sparsely annotated proteins and will systematically misclassify minority-class variants of proteins for which both deleterious and non-deleterious variants are seen at inference time[11].

In this work, we address the problem of information leakage due to misleading evaluation on in-distribution data for models intended for deployment on OOD data. To this end, we developed DataSAIL, an algorithmic framework and tool to split datasets into multiple folds that allow us to realistically estimate model performance on OOD data. While such datasets have been curated for specific ML tasks on biomolecular data (e.g., PINDER[13] and the gold standard dataset developed in[10] for PPI prediction and PLINDER[14] for protein-ligand interaction prediction), DataSAIL is generic and can be used to split any kind of data, as long as a similarity or distance measure for the contained data points is available.

We formulate the data splitting problem underlying DataSAIL as a constrained optimization problem, prove that this problem is NP-hard, and present a Python package that heuristically solves this problem using clustering and integer linear programming (ILP). Unlike existing tools and algorithms to compute data splits that reduce information leakage, DataSAIL can automatically compute splits for heterogeneous data of two different types and combines stratification with similarity-aware splitting (see Table 1 and section "Detailed description of related work" in the Supplementary Materials for comparison with existing tools). Moreover, DataSAIL is more versatile than existing approaches because it can be used out-of-the-box for various types of molecular data. We validate DataSAIL by showing how it can reduce leakage between training and test data for various ML models trained on both one- and two-dimensional biomolecular datasets.

## Results

### Data splits for supervised ML
In supervised ML, we are given a dataset $\mathcal{M} = \{(x_1, y_1), \ldots, (x_n, y_n)\}$ of $n$ samples with feature vectors $\mathbf{x_i} \in X$ and labels $y_i \in Y$, where $X$ is a feature space and $Y$ represents the space of labels. The goal is to learn a function $f_\theta: X \to Y$ that minimizes a loss function $\mathcal{L}(f_\theta(\mathbf{x_i}), y_i)$. This is achieved by selecting a hypothesis space $\mathcal{H}$ of candidate functions and fitting $f_\theta \in \mathcal{H}$ within the hypothesis space[15]. To develop a supervised ML model $f_\theta$, one needs to split $\mathcal{M}$ into three pairwise disjoint datasets: A training set $\mathcal{M}_{train}$ to learn the parameters $\theta$, i.e., to select $f_\theta$ from a fixed hypothesis space $\mathcal{H}$. A validation set $\mathcal{M}_{val}$ to optimize the hyperparameters that determine the shape of $\mathcal{H}$ (e.g., number of hidden layers) or control the employed optimization strategy (e.g., learning rate or optimizer). And a test set $\mathcal{M}_{test}$ to assess the performance of the trained model on so far unseen data.

Our proposed method, DataSAIL, works for one-dimensional and two-dimensional datasets. In a one-dimensional dataset, one feature vector-output value pair $(\mathbf{x_i}, y_i)$ corresponds to one elementary data point; for example, a certain molecular property such as toxicity can be predicted for a single chemical compound (Fig. 1, 1D data). If the feature vector $\mathbf{x_i}$ consists of two elementary data points (such as in drug-target interaction prediction, where $\mathbf{x_i}$ represents a pair of a molecule and a protein target for which an interaction affinity $y_i$ should be predicted), we call this a two-dimensional dataset (Fig. 1, 2D data).

Importantly, in a two-dimensional dataset, the similarity between molecules can be defined over each dimension, e.g., along the drug and target dimensions. We define different splitting tasks with abbreviations based on whether they account for similarity-induced information leakage and the dimensions of the dataset (1 or 2). In identity-based splittings, the similarity between molecules is not considered, whereas in similarity-based splittings, it is accounted for. Those tasks are visualized in Fig. 1 and include identity-based one-dimensional splitting (I1), identity-based two-dimensional splitting (I2), similarity-based one-dimensional splitting (S1), similarity-based two-dimensional splitting (S2), and random interaction-based splitting (R). In two-dimensional data splitting, interactions may exist that cannot be assigned to any split without leaking information if the two interacting molecules are assigned to different folds. Therefore, interactions can get lost in two-dimensional data splitting (white tiles in panels I2 and S2 in Fig. 1).

### The ($k$, $R$, $C$)-DataSAIL problem
In this section, we introduce ($k$, $R$, $C$)-DataSAIL, which formalizes the problem of splitting an $R$-dimensional dataset into $k$ folds such that data leakage is minimized and $C$ classes that are present in the data (e.g., confounders such as sex or the output labels $y_i$ if the space of labels $Y$ is discrete) are distributed equally among the $k$ folds such that each fold preserves the overall class distribution. Intuitively, we define ($k$, $R$, $C$)-DataSAIL as the problem to minimize inter-class similarity while keeping similar class ratios across the splits. Although designed with biomedical applications in mind, our problem definition is generic and can be used for any dataset where a similarity or a distance measure is available for the contained data points. We present a

## Table 1 | Overview of different splitting tools and frameworks and their capabilities towards biochemical data

| Tool | TDC[46] | DeepChem[26] | sklearn[32] | LoHi[27] | GraphPart[28] | astartes[47] | DataSAIL |
|---|---|---|---|---|---|---|---|
| Features | | | | | | | |
| 1D splits | ✓ | ✓ | ✓ | ✓ | ✓ | ✓ | ✓ |
| 2D splits | ✗ | ✗ | ✗ | ✗ | ✗ | ✗ | ✓ |
| Stratified splits | ✓ | ✓ | ✓ | ✗ | ✗ | ✗ | ✓ |
| Preserves all data (1D splits) | ✓ | ✓ | ✓ | ✗ | ✗ | ✓ | ✓ |
| Preserves all data (2D splits) | N/A | N/A | N/A | N/A | N/A | N/A | ✗ |
| Supported input data | | | | | | | |
| Proteins | ✗ | ✗ | ✗ | ✗ | ✓ | ✗ | ✓ |
| Small molecules | ✓ | ✓ | ✗ | ✓ | ✗ | ✓ | ✓ |
| DNA & RNA sequences | ✗ | ✗ | ✗ | ✗ | ✗ | ✗ | ✓ |
| Genomes & longer contigs | ✗ | ✗ | ✗ | ✗ | ✗ | ✗ | ✓ |
| Custom data | ✗ | ✗ | ✗ | ✗ | ✗ | ✗ | ✓ |

Checkmarks indicate that a tool can compute a split with the requested property without requiring preprocessing by the user.

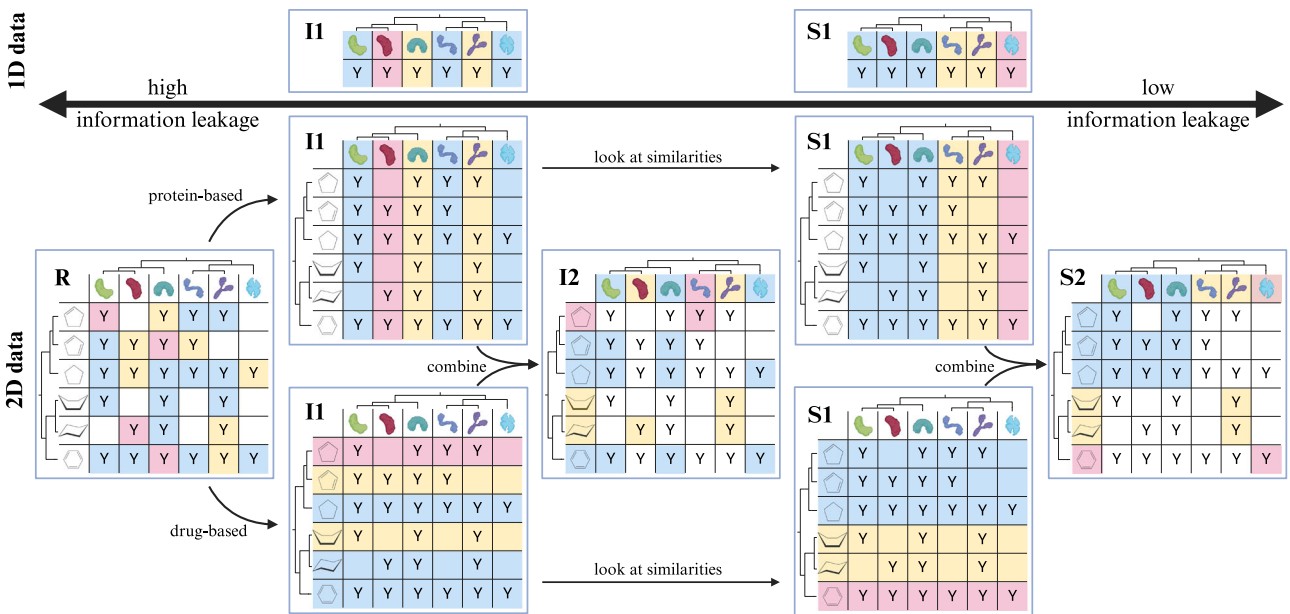

**Fig. 1 | Visualization of exemplary one-dimensional and two-dimensional datasets.** The symbol "Y" indicates the presence of a measurement, while the phylogenetic trees next to the matrix visualizations illustrate similarities between samples. The figure showcases all splitting tasks and their interrelations. Samples assigned to training are highlighted with a blue background, validation samples are in yellow, and test samples are marked in red. Unassignable tiles are left white. Created in BioRender. Joeres, R. (2025) https://BioRender.com/w47j283.

generalized version of the problem for $R$-dimensional datasets, although our Python implementation only supports one- or two-dimensional input, i.e., $R \leq 2$.

More formally, let $\mathcal{D}$ be the set of data points represented in $\mathcal{M}$ or a set of clusters defined over these data points (allowing clusters as elements of $\mathcal{D}$ will be important for our heuristic solver, as explained below). The data points/clusters $x \in \mathcal{D}$ can have $R \in \mathbb{N}$ different entity types $t(x) \in [R] := \{1, \ldots, R\}$. For instance, in a drug-target interaction scenario, we have $R = 2$, with the two entity types $r \in \{1, 2\}$ corresponding to drugs and protein targets or clusters thereof. If $\mathcal{M}$ is a one-dimensional dataset, all data points/clusters have the same element type. Moreover, the data points or clusters $x \in \mathcal{D}$ have cardinalities $\kappa(x) \in \mathbb{N}_{\geq 1}$. For elementary data points, we always have $\kappa(x) = 1$; if $\mathcal{D}$ contains clusters, we may have $\kappa(x) > 1$. For each element type $r \in [R]$, we write $\mathcal{D}_{t=r} := \{x \in \mathcal{D} | t(x) = r\}$ and $n_r := \sum_{x \in \mathcal{D}_{t=r}} \kappa(x)$ to denote the set of all data elements of type $r$ and their overall cardinality, respectively. Additionally, we assume that a similarity measure $\text{sim} : \mathcal{D} \times \mathcal{D} \to \mathbb{R}$ or a distance measure $\text{dist} : \mathcal{D} \times \mathcal{D} \to \mathbb{R}$ is available for $\mathcal{D}$ (see "Methods" for details on how to define sim and dist).

Information leakage has been defined qualitatively by Kaufman et al.[3] and quantitatively by Elangovan et al.[16] as follows:

$$|\mathcal{D}_{test}|^{-1} \cdot \sum_{x \in \mathcal{D}_{test}} \max_{x' \in \mathcal{D}_{train}} \text{sim}(x, x'). \tag{1}$$

This definition is incomplete as only the biggest leak per test sample is considered, and the validation set is ignored. In view of this, we define the leakage induced by a mapping $\pi : \mathcal{D} \to [k]$ that splits $\mathcal{D}$ into $k$ folds $\mathcal{D}_i^\pi := \{x \in \mathcal{D} | \pi(x) = i\}$, $i \in [k] := \{1, \ldots, k\}$, as the total similarity

$$L(\pi) := \sum_{xx' \in \binom{\mathcal{D}}{2}} [\pi(x) \neq \pi(x')] \cdot \text{sim}(x, x') \cdot \kappa(x) \cdot \kappa(x') \tag{2}$$

between data elements assigned to different folds. Here, $[\cdot] : \{\bot, \top\} \to \{0, 1\}$ is the Iverson bracket. The cardinalities are added as factors to our leakage function $L$ to put a higher weight on similarities between larger clusters. Typically, we have $k = 3$ for data splitting in ML ($\mathcal{D}_1^\pi = \mathcal{D}_{train}$, $\mathcal{D}_2^\pi = \mathcal{D}_{val}$, $\mathcal{D}_3^\pi = \mathcal{D}_{test}$). We will define $(k, R, C)$-DataSAIL as the problem to minimize $L(\pi)$, given two sets of constraints we introduce below.

Let $s_i \in (0, 1)$ with $\sum_{i=1}^k s_i = 1$ be user-provided desired split fractions for the $k$ folds $\mathcal{D}_i^\pi$ induced by $\pi$ (e.g., $s_1 = 0.8$, $s_2 = 0.1$, $s_3 = 0.1$ for splitting the data into 80% training, 10% validation, and 10% test data). As a first set of constraints on $\pi$, we require that, for all pairs $(i, r) \in [k] \times [R]$ of entity types and folds, $\pi$ respects the split fractions $s_i$ up to a relative error $\epsilon \in [0, 1)$:

$$\sum_{x \in \mathcal{D}_i^\pi \cap \mathcal{D}_{t=r}} \kappa(x) \geq (1 - \epsilon) \cdot s_i \cdot n_r \tag{3}$$

For elementary data points where $\kappa(x) = 1$, this means that the fraction of data points of type $r$ (that is, the data points in $\mathcal{D}_{t=r}$) that are assigned to split $i$ (the data points in $\mathcal{D}_i^\pi$) matches the desired split fractions up to a relative error $\epsilon$.

In many ML applications, the data elements $x \in \mathcal{D}$ may belong to one or multiple of $C$ classes $\sigma(x) \subseteq [C]$, and we would like to compute stratified splits where the desired split fractions $s_i$ are respected for each class $c \in C$. To model this requirement, we add a constraint

$$\sum_{x \in \mathcal{D}_i^\pi \cap \mathcal{D}_{t=r}^{\sigma=c}} \kappa(x) \geq (1 - \delta) \cdot s_i \cdot n_r^c \tag{4}$$

for each triple $(i, r, c) \in [k] \times [R] \times [C]$ of folds, entity types, and classes, where $\mathcal{D}_{t=r}^{\sigma=c} := \{x \in \mathcal{D}_{t=r} | c \in \sigma(x)\}$ is the set of data elements of type $r$ that belong to class $c$, $n_r^c := \sum_{x \in \mathcal{D}_{t=r}^{\sigma=c}} \kappa(x)$ is the overall cardinality of such data elements, and $\delta \in [0, 1]$ is an acceptable relative error. Two observations are important at this point:

- For all $\epsilon \geq \delta$, the set of constraints specified in Eq. (4) implies the constraints from Eq. (3), which can thus be discarded if $\delta \leq \epsilon$.
- When no class information is available (i.e., all data elements have the same "dummy class" $\sigma(x) = 1$), Eqs. (4) and (3) are equivalent up to choices of $\epsilon$ and $\delta$ and Eq. (4) can thus be discarded.

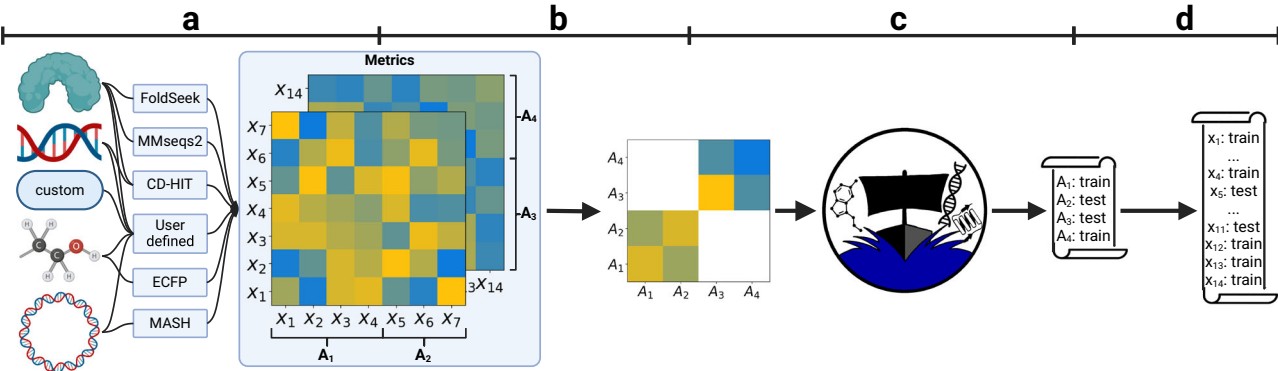

**Fig. 2 | Schematic workflow of DataSAIL.** Input can be any data (special focus is on biochemical data). DataSAIL then computes a pairwise distance or similarity matrix (**a**) and stratifies the data into a constant number of clusters based on that (**b**). These clusters are split into $k$ folds, using off-the-shelf ILP solvers (**c**). From the partitioning of the clusters, DataSAIL infers the partitioning of the elementary data points (**d**). Created in BioRender. Joeres, R. (2025) https://BioRender.com/m81k197.

We can now define the $(k, R, C)$-DataSAIL problem:

$$\min_{\pi} L(\pi) \qquad (5)$$

$$\text{s.t.} (3), (4) \qquad (6)$$

**Theorem 1.** The $(k, R, C)$-DataSAIL problem is NP-hard for all $k \in \mathbb{N}_{\geq 2}$, $R \in \mathbb{R}_{\geq 1}$, and $C \in \mathbb{N}_{\geq 1}$.

The proof for Theorem 1 is contained in Section "Proof of Theorem 1". To compute leakage-reduced data splits despite this hardness result, we developed a heuristic workflow that first assigns individual data points to a fixed number of clusters and then solves a constant-size instance of $(k, R, C)$-DataSAIL where the clusters are treated as data elements (Fig. 2, see "Methods" for details). The $(k, R, C)$-DataSAIL problem can be formulated as an ILP with $\mathcal{O}(|\mathcal{D}| \cdot k + |\mathcal{D}|^2)$ variables and $\mathcal{O}(k \cdot R \cdot C + |\mathcal{D}|^2 \cdot k)$ constraints. The formulation of the ILP problem is given in Section "An ILP formulation of the $(r, R, C)$-DataSAIL problem". Note that when $|\mathcal{D}|$ is a constant-sized set of pre-computed clusters over the original dataset, our ILP has a constant number of variables and constraints and can hence be solved efficiently. To solve the constant-size instances, we use the standard ILP solvers.

Once a mapping $\pi$ has been computed for $\mathcal{D}$, we can use it to compute a mapping for $\mathcal{M}$: First, we unpack $\pi$ and assign each data element $z$ contained in the cluster $x \in \mathcal{D}$ to $\pi(x)$, i.e., we define $\pi(z) := \pi(x)$ for all $z \in x$ (Fig. 2d). Then, we assign the feature vector-label pair $(x_j, y_j) \in \mathcal{M}$ to the split $i$ if and only if all $\pi(z) = i$ holds for all data points $z$ represented by $x_j$ (e.g., a drug and a protein in the case of drug-target interaction prediction). Feature vector-label pairs $(x_j, y_j)$ with conflicting assignments for different data points represented by $x_j$ are discarded. For instance, in a drug-target prediction scenario, it may happen that the drug and the protein jointly represented by $x_j$ are assigned to different splits by $\pi$ (e.g., the drug is assigned to the training fold and the protein is assigned to the test fold). In this case, $(x_j, y_j)$ is discarded.

**Splitting biomolecular datasets**

First, we consider one-dimensional data. We trained and tested four baseline ML models (random forests[17] (RF), support vector machines[18] (SVM), gradient boosting[19] (XGB), and multilayer perceptrons[20] (MLP)) and the deep learning model D-MPNN[21] for molecular property predictions on random and similarity-based data splits computed with DataSAIL (S1) and two competitors. We used two widely used datasets from the MoleculeNet collection[22] (QM8: regression problem, upper panels in Fig. 3; Tox21: classification problem, lower panels; Supplementary Fig. 1 shows further results for additional competitors and

further one-dimensional datasets from MoleculeNet). As expected, splitting with DataSAIL leads to a better separation of training and test samples. In particular, on both datasets, DataSAIL's data splits exhibit the lowest leakage $L(\pi)$ among all compared data splits (see rightmost bar-plots in Fig. 3c, f). The other tools that aim to reduce information leakage—LoHi and DeepChem's fingerprint-based splitting—only partly achieve this goal, thus leading to larger values of $L(\pi)$. Overall, we observe that smaller values of $L(\pi)$ are associated with larger drops in test performance in comparison to random splits (see Fig. 3c, f and Supplementary Fig. 2). This indicates that minimizing $L(\pi)$ as implemented in DataSAIL indeed leads to harder splits and also shows that the ML models benchmarked here struggle to generalize to molecules with low Tanimoto similarity[23–25] (the similarity measure we used to compute the values of $L(\pi)$ reported in Fig. 3) with respect to the training data. For the deep learning model D-MPNN, these results are in line with findings reported in the original publication[21], where the authors had shown that D-MPNN performs substantially worse on scaffold-based splits (which, like DataSAIL's S1 splits, rely on molecular similarity) than on random splits (see Supplementary Table 1). In all figures, we depict the scaled $L(\pi)$ as defined in Eq. (20).

Then, we consider two-dimensional data by splitting the LP-PDBBind dataset that contains binding affinities between 15,477 drugs and 12,718 protein targets (Fig. 4). We compared DataSAIL's I2 and S2 splitting to I1 and S1 splitting for both drugs and targets, as well as to DeepChem's fingerprint-based splitting[26], LoHi[27], and GraphPart[28] (comparisons to additional splitting algorithms are shown in Supplementary Figs. 3 and 4). As for the one-dimensional data, splits computed by DataSAIL exhibit consistently low $L(\pi)$ values (Fig. 4c, f, i), with the S2 splits performing particularly well. Moreover, splits with low $L(\pi)$ values again lead to substantial drops in performance in comparison to I1 baselines which split the data randomly across the drug (Fig. 4c) or protein (Fig. 4f) axis. Another interesting observation is that, for all ML models, test performances are substantially worse for DataSAIL's S2 splits than for all other tested data splits (Fig. 4i), showing that the tested binding affinity prediction models do not generalize well to scenarios where neither the drugs nor the proteins seen at inference time are similar to drugs and proteins contained in the training data. To our knowledge, DataSAIL is the only tool with out-of-the-box support for a splitting strategy that allows for the estimation of generalization capability in such scenarios. Strikingly, a comparison with the dataset-specific splits in the protein-ligand dataset PLINDER (Supplementary Table 2) shows that, in terms of $L(\pi)$, Data-SAIL's automatically computed splits are competitive with splits curated for specific datasets.

Another improvement of DataSAIL over existing methods is the combination of stratified splitting with information leakage minimization. To show the effect of DataSAIL in this setting, we use the

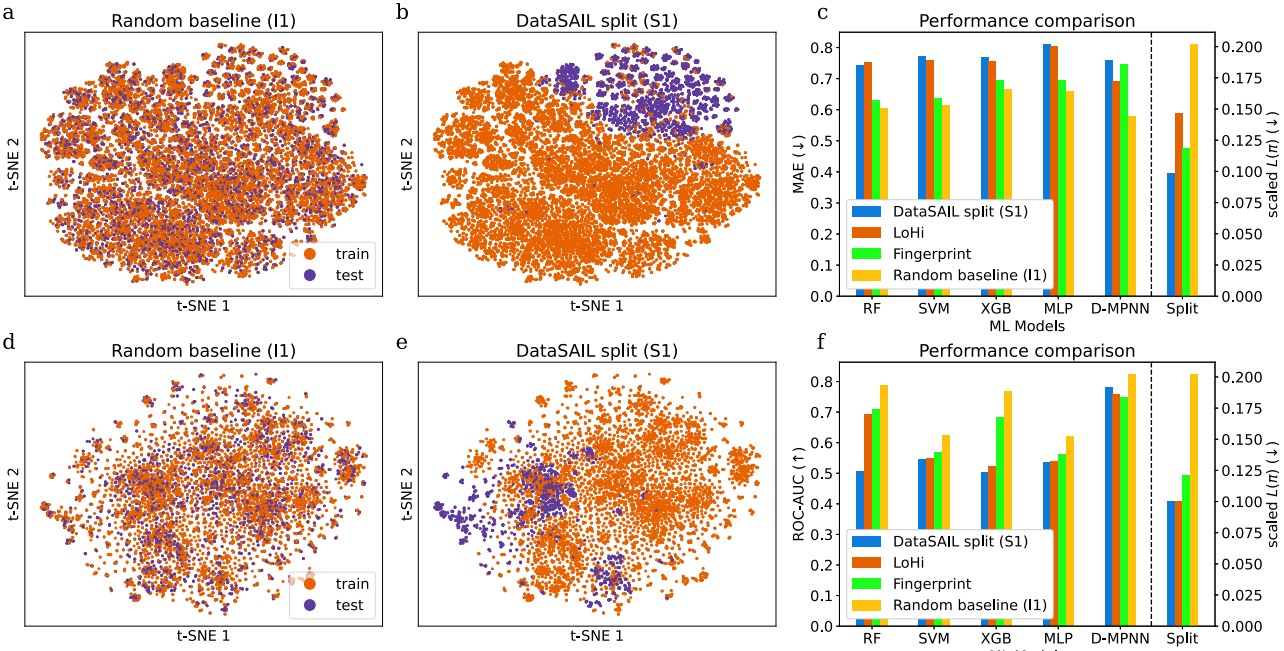

**Fig. 3 | One-dimensional datasets QM8 and Tox21.** We show QM8 (**a**, **c**, **e**) and Tox21 (**b**, **d**, **f**) from the MoleculeNet benchmark collection. **a**, **d** show the t-SNE embeddings for random split and **b**, **e** for DataSAIL's S1 split. **c**, **f** show ML model performances and information leakages for the different splits, quantified using mean absolute errors (MAE, lower is better) for QM8 and area under the receiver operating characteristic (ROC-AUC, higher is better) for Tox21.

SR-ARE subchallenge from Tox21, for which 6889 active and 942 inactive small molecules exist in the dataset. Here, DataSAIL is not compared to a fully random split but to the classical, similarity-unaware stratified split. We investigate the effect of additionally introducing similarity-awareness and observe that the corresponding DataSAIL splits reduce information leakage considerably (Fig. 5). Again, the comparison of $L(\pi)$ on the right of Fig. 5c shows that DataSAIL computes splits with reduced information leakage between the folds in comparison to the classical methods, and again we observe that these splits pose harder generalization tasks, leading consistent drops in performance across all tested ML models.

### Effect of solvers and hyper-parameters, scalability

An important parameter of DataSAIL is the number of clusters $K$ used to construct the constant-size $(k, R, C)$-DataSAIL instance to be fed into the ILP solver. The first row of Fig. 6 shows how $K$ affects the quality of the splits (panel a) and the runtime of DataSAIL (panel b). We clustered the Tox21 dataset into various numbers of clusters using Tanimoto similarities of ECFPs. We then fed the resulting $(k, R, C)$-DataSAIL instances into the ILP solvers GUROBI, MOSEK, and SCIP and set a time limit of 2 h per solver. Interestingly, the quality of the splits does not improve for $K > 150$ and is already good for $K \approx 50$, showing that a rather small number of clusters is sufficient for obtaining leakage-reduced data splits with DataSAIL. In terms of quality, the tested ILP solvers perform very similarly. GUROBI is the fastest solver.

Figure 6c shows how the quality of the splits depends on the acceptable relative errors $\epsilon$ and $\delta$, tested for the SR-ARE subchallenge from Tox21 with quality quantified following Eq. (2). The classes of this dataset were the binary labels of the SR-ARE subchallenge. Therefore, we balanced positive and negative samples in both splits. We observe that the quality mainly depends on $\epsilon$, which controls how close the obtained split fractions have to be to the user-requested split fractions $s_i$. Contrary to expected, we did not identify a dependency on $\delta$. However, this is only a small example, and general trends may differ as datasets can vary greatly.

Because MoleculeNet offers a variety of datasets with different sizes, structures, and similarities, we use it for benchmarking the runtime of the various splitting techniques from DataSAIL, LoHi[27], and DeepChem[26] (Fig. 6d). As expected, the bigger the dataset, the slower the algorithms compute their splits. While DataSAIL is the slowest algorithm, it shows a benign scaling behavior and terminated for all datasets within a reasonable amount of time. In contrast, LoHi did not produce results for the MUV dataset within 12 h.

## Discussion

Similarity is an often overlooked source of information leakage that is especially relevant when ML models are developed to be used on data with a distribution shift during inference. In this work, we present DataSAIL, a computational workflow and a tool to minimize similarity-induced information leakage when splitting data for ML model training and testing. DataSAIL provides better OOD data splits than state-of-the-art tools. We provide a formal definition of the underlying optimization problem, show that the problem is NP-hard, present a scalable heuristic, and empirically show that our heuristic can compute high-quality leakage-reduced data splits in a reasonable time, making DataSAIL a Swiss army knife for data splitting. DataSAIL can split one-dimensional and two-dimensional data and biochemical data of various types (small molecules, protein sequences, DNA and RNA sequences, genomes, and longer contigs). Our framework can also easily accommodate other data types, provided the user can provide similarities or distances between the data points.

A limitation of our implementation of DataSAIL is that it only supports $R \leq 2$ entity types, although our theoretical framework applies to arbitrary $R$-dimensional data. In future work, we plan to extend the DataSAIL implementation to work on arbitrary dimensional data. Another current limitation is the clustering step (Fig. 2b), where our implementation relies on spectral or agglomerative clustering and does not support custom clustering algorithms that may be more appropriate for specific data types. DataSAIL's implementation also cannot handle similarities between entities of different types, although the theoretical framework allows for that. Moreover, splitting two-dimensional data with the current version of DataSAIL can lead to the loss of some feature vector-label tuples when the two elementary data points represented by the feature vector are assigned to different

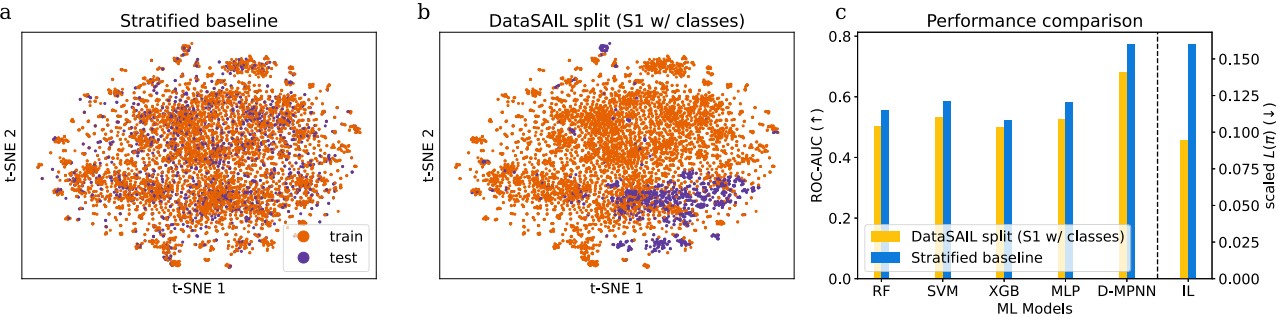

**Fig. 4 | Two-dimensional dataset LP-PDBBind. a**, **d** show the t-SNE embeddings for random splits (I1), **b**, **e** for one-dimensional similarity-based splits (S1), and **g**, **h** for two-dimensional similarity-based splits (S2). **a**, **b**, **g** show t-SNE embeddings of the drugs' ECFP4 fingerprints; t-SNE embeddings of the proteins' ESM2-t12 embeddings[53] are visualized in (**d**, **e**, **h**) (gray dots visualize data points that had to be dropped for the two-dimensional splits). **c**, **f**, **i** show ML model performances, measured via the root mean squared error (RMSE, lower is better) and information leakages for the different splits.

**Fig. 5 | One-dimensional data splitting with stratification.** We used the SR-ARE target dataset in Tox21, where the two classes are active and inactive small molecules. **a** t-SNE embeddings for a random stratified split, **b** for a similarity-aware stratified split. **c** ML model performances and information leakage for the different splits.

splits. This problem could be mitigated by adding a data loss penalization term to the objective function minimized by DataSAIL. Expanding the implementation to the theoretical limits would improve the versatility of the Python package but also increase the number of variables to deal with. Furthermore, DataSAIL uses off-the-shelf ILP solvers that naturally have a high overhead because they are applicable to multiple settings. Tailoring a solver to DataSAIL's specific needs may thus improve performance and runtime considerably.

Finally, it is important to stress that testing models on challenging OOD splits as computed by DataSAIL is not appropriate in every ML development setting: The leakage function $L(\pi)$ minimized by Data-SAIL becomes small when $\text{sim}(x, x')$ is small for data elements $x$ and $x'$ that $\pi$ assigns to different splits. Since DataSAIL allows the user to select from various pre-implemented similarity functions sim and is open to custom similarity functions, the user has full control over the behavior of $L(\pi)$. Which choice of sim is most appropriate depends on

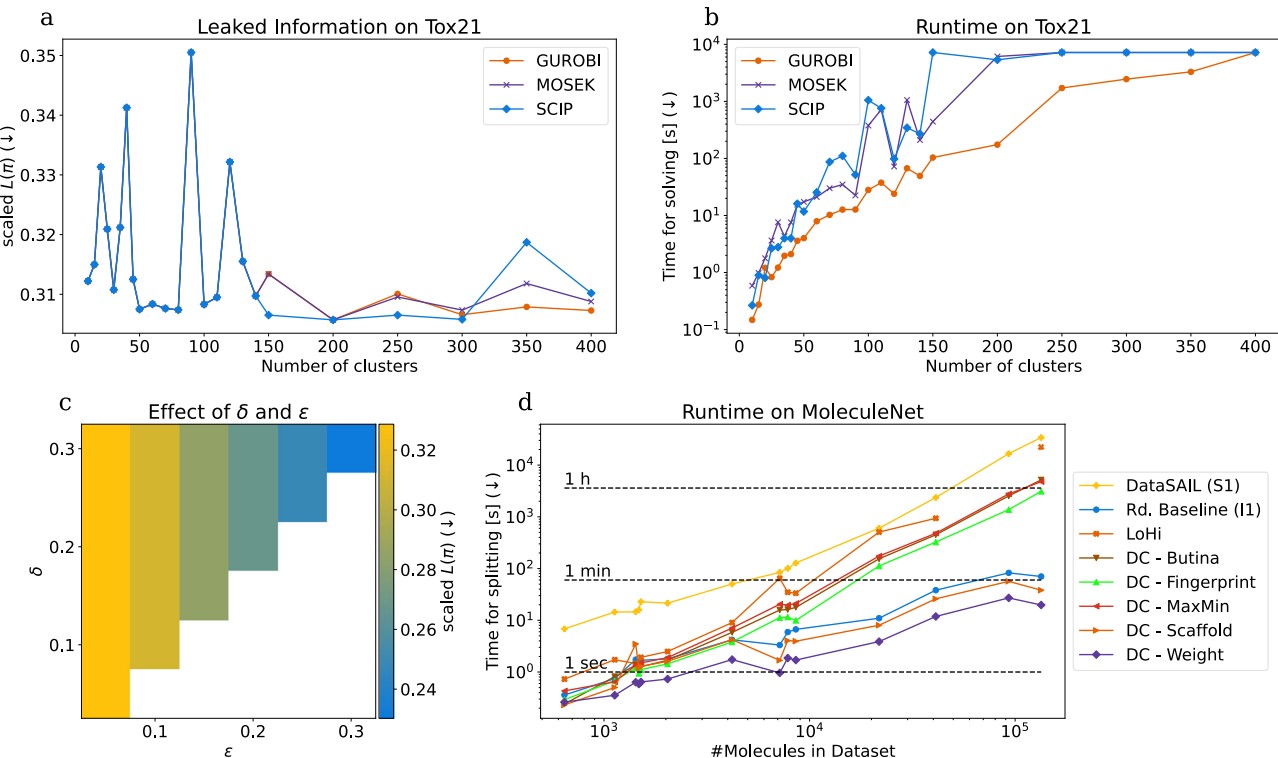

**Fig. 6 | Ablation studies and scalability benchmark.** Quality of DataSAIL splits (**a**) and runtime (**b**) as a function of the numbers of clusters $K$. **c** Effect of acceptable error margins $\epsilon$ and $\delta$ on split quality. **d** Runtimes of DataSAIL and other tools as a function of dataset size. In the legend, DC abbreviates DeepChem.

the intended deployment scenario for the evaluated ML model. In particular, if the inference-time data is expected to be similar to the training data with respect to the similarity function sim selected by the user, evaluating an ML model on the data splits computed by DataSAIL will lead to overly pessimistic results. When using DataSAIL to compute splits for evaluating an ML model that is intended to generalize to OOD data, model evaluators hence have to ensure that the selected similarity function sim indeed captures the intended generalization task. Given an appropriate choice of sim, a positive correlation between $L(\pi)$ and performance then indicates that the tested ML models struggle to generalize the OOD scenarios modeled by sim.

Related to this, it may happen that the selected similarity function sim is correlated with the response variable to be predicted by the ML model (e.g., in a binary classification problem, it could happen that $\text{sim}(x, x')$ is substantially larger for data points $x$ and $x'$ that fall into the same class than for data points that fall into different classes). In such scenarios, it is crucial that the user runs DataSAIL with the stratification constraint (4), where the classes $C$ are defined according to the response variable. Without such a constraint, DataSAIL would compute splits that are highly imbalanced with respect to the response variable, which may again lead to overly pessimistic performance estimates.

## Methods
### Proof of Theorem 1
We show that the $(k, R, C)$-DataSAIL problem is NP-hard for all fixed constants $k \in \mathbb{N}_{\geq 2}$ (number of folds), $R \in \mathbb{N}_{\geq 1}$ (number of entity types), and $C \in \mathbb{N}_{\geq 1}$ (number of classes). We proceed in three steps:
- Step 1: We show that there is a polynomial-time reduction from $(k, R, 1)$-DataSAIL to $(k, R, C)$-DataSAIL for arbitrary fixed $C \geq 2$.
- Step 2: We show that there is a polynomial-time reduction from $(k, 1, 1)$-DataSAIL to $(k, R, 1)$-DataSAIL for arbitrary fixed $R \geq 2$.
- Step 3: We show that $(k, 1, 1)$-DataSAIL is NP-hard via a polynomial-time reduction from the minimum $k$-section problem, which is known to be NP-hard.

Step 1 is straightforward: Given an instance $I_{k,R,1} = (\mathcal{D}, \text{sim}, \kappa, \{s_i\}_{i=1}^k, \epsilon)$ of $(k, R, 1)$-DataSAIL (we can ignore the $\delta$ if $C = 1$), we construct an instance $I_{k,R,C}$ of $(k, R, C)$-DataSAIL by arbitrarily assigning the data elements $x \in \mathcal{D}$ to $C$ classes and setting $\delta := 1$. Then, Eq. (4) is vacuous, implying that each $\pi : \mathcal{D} \to [k]$ is a solution to $I_{k,R,1}$ if and only if it is a solution to $I_{k,R,C}$.

For Step 2, let $I_{k,1,1} = (\mathcal{D}_1, \text{sim}, \kappa, \{s_i\}_{i=1}^k, \epsilon)$ of $(k, 1, 1)$-DataSAIL. We now construct and instance $I_{k,R,1} = (\mathcal{D}', \text{sim}', \kappa', \{s_i\}_{i=1}^k, \epsilon)$ of $(k, R, 1)$-DataSAIL as follows: $\mathcal{D}'$ contains $\mathcal{D}_1$ and $R - 1$ additional copies $\mathcal{D}_r$, $r = 2, ..., R$. Let $x_r$ denote the copy in $\mathcal{D}_r$ of the data element $x_1 \in \mathcal{D}_1$. We define $\kappa'(x_r) := \kappa(x_1)$ for all copies. For all pairs of data elements $(x_r, x'_{r'}) \in \mathcal{D}' \times \mathcal{D}'$, we define

$$\text{sim}'(x_r, x'_{r'}) := \begin{cases} M & \text{if } r \neq r' \text{ and } x_1 = x'_1, \\ \text{sim}(x_1, x'_1) & \text{otherwise} \end{cases}, \quad (7)$$

where $M$ is some large enough constant ($M := R^2 \cdot \sum_{x_1 x'_1 \in \binom{\mathcal{D}_1}{2}} \text{sim}(x_1, x'_1)$ suffices). That is, similarities between different copies of the same data element are set to a very high value $M$, and all other similarities are inherited from the $(k, 1, 1)$-DataSAIL instance $I_{k,1,1}$.

Given an optimal solution $\pi_1$ for $I_{k,1,1}$, we can always define an induced solution $\pi_R$ for $I_{k,R,1}$ as $\pi_R(x_r) := \pi_1(x_1)$. Eq. (3) continues to hold because we have $\kappa'(x_r) = \kappa(x_1)$ for all $r \in [R]$ and $x_1 \in \mathcal{D}_1$. For each edge $x_1 x'_1$ contained in the cut induced by $\pi_1$, there are $\binom{R}{2} + R$ copies contained in the cut induced by $\pi_R$, all of which have weight $\text{sim}(x_1, x'_1) \cdot \kappa(x_1) \cdot \kappa(x'_1)$: $R$ copies of the form $x_r x'_r$ and $\binom{R}{2}$ copies of the form $x_r x'_{r'}$ with $r \neq r'$. Moreover, the cut contains no other edges since, by definition of $\pi_R$, all copies of the same node end up in the

same split. Hence, we have

$$OPT_R \leq L(\pi_R) = \left( \binom{R}{2} + R \right) \cdot L(\pi_1) = \left( \binom{R}{2} + R \right) \cdot OPT_1 < M, \quad (8)$$

where $OPT_1$ and $OPT_R$ denote the optima of $I_{k,1,1}$ and $I_{k,R,1}$, respectively.

Conversely, let $\pi'_R$ be an optimal solution for $I_{k,R,1}$. Then $\pi'_R$ puts all copies of the same data elements into the same folds, since otherwise, we would have $L(\pi'_R) \geq M > L(\pi_R)$, contradicting the optimality of $\pi'_R$. For all $x_1 \in \mathcal{D}_1$, we now define $\pi'_1(x_1) := \pi'_R(x_1)$. By counting edge copies as above, we obtain:

$$OPT_R = L(\pi'_R) = \left( \binom{R}{2} + R \right) \cdot L(\pi'_1) \geq \left( \binom{R}{2} + R \right) \cdot OPT_1. \quad (9)$$

By combining the chains of inequalities in Eqs. (8) and (9), we obtain that $\pi'_1$ is optimal for $I_{k,1,1}$. This concludes Step 2 of our proof.

For Step 3, we have to show that $(k, 1, 1)$-DataSAIL is NP-hard for all constants $k \in \mathbb{N}_{\geq 2}$. This can be done via a reduction from the minimum $k$-section problem. Given a graph on $G = (V, E)$ and a constant $k \in \mathbb{N}_{\geq 2}$, the minimum $k$-section problem asks to find a partition $\pi: V \to [k]$ that splits $V$ into $k$ folds such that $\sum_{uv \in E} [\pi(u) \neq \pi(v)]$ is minimized and

$$\left\lfloor \frac{|V|}{k} \right\rfloor \leq |V_i^\pi| \leq \left\lceil \frac{|V|}{k} \right\rceil \quad (10)$$

holds for all $i \in [k]$ (all folds have the same size). This problem is NP-hard, even when restricting to balanced instances with $|V| = k \cdot C$ for some $C \in \mathbb{N}_{\geq 1}$[29], where the constraint in Eq. (10) simplifies to

$$|V_i^\pi| = k^{-1} \cdot |V|. \quad (11)$$

Given a balanced instance $(V, E, k)$ of the minimum $k$-section problem, we now define an instance $I_{k,1,1} = (\mathcal{D}, sim, \kappa, \{s_i\}_{i=1}^k, \epsilon)$ of $(k, 1, 1)$-DataSAIL by setting, $\mathcal{D} := V, \kappa(x) := 1$ for all $x \in \mathcal{D}$, $sim(x, x') := [xx' \in E]$ for all $(x, x') \in \mathcal{D} \times \mathcal{D}$, $s_i := k^{-1}$ for all $i \in [k]$, and $\epsilon := 0$. Clearly, any solution $\pi$ to $(V, E, k)$ also solves $I_{k,1,1}$ and vice versa. Moreover, we have $L(\pi) = \sum_{uv \in E} [\pi(u) \neq \pi(v)]$ by construction of sim. Consequently, solving $(V, E, k)$ is equivalent to solving $I_{k,1,1}$, which completes the proof.

### An integer linear programming formulation of the $(r, R, C)$-DataSAIL problem

Our formulation contains binary variables $\xi_{x,i}$ for all $(x, i) \in \mathcal{D} \times [k]$ that encode whether the data element $x$ is assigned to fold $i$. Moreover, it contains binary variables $\zeta_{xx'}$ for all unordered pairs of data elements $xx' \in \binom{\mathcal{D}}{2}$, which are defined such that $\zeta_{x,x'} = 1$ if and only if $x$ and $x'$ are assigned to different folds.

$$\min_{\xi, \zeta} \sum_{xx' \in \binom{\mathcal{D}}{2}} sim(x, x') \cdot \zeta_{xx'} \quad (12)$$

$$\sum_{i=1}^k \xi_{x,i} = 1 \quad \forall x \in \mathcal{D} \quad (13)$$

$$\sum_{x \in \mathcal{D}_{t=r}} \kappa(x) \cdot \xi_{x,i} \geq (1 - \epsilon) \cdot s_i \cdot n_r \quad \forall (i, r) \in [k] \times [R] \quad (14)$$

$$\sum_{x \in \mathcal{D}_{t=r}^{a=c}} \kappa(x) \cdot \xi_{x,i} \geq (1 - \delta) \cdot s_i \cdot n_r^c \quad \forall (i, r, c) \in [k] \times [R] \times [C] \quad (15)$$

$$\zeta_{xx'} \geq \xi_{x,i} - \xi_{x',i} \quad \forall (xx', i) \in \binom{\mathcal{D}}{2} \times [k] \quad (16)$$

$$\xi_{x,i} \in \{0, 1\} \quad \forall (x, i) \in \mathcal{D} \times [k] \quad (17)$$

$$\zeta_{xx'} \in \{0, 1\} \quad \forall xx' \in \binom{\mathcal{D}}{2} \quad (18)$$

Constraint (13) ensures that $\boldsymbol{\xi}$ encodes a partition $\pi_\xi$. Constraints (14) and (14) ensure that $\pi_\xi$ respects the constraints from Eqs. (3) and (4), respectively. Constraint (16) ensures that

$$\zeta_{xx'} = [\pi_\xi(x) \neq \pi_\xi(x')], \quad (19)$$

which implies that the objective minimized in (12) equals $L(\pi_\xi)$. This, in turn, implies that the ILP given in the equations (12) to (18) is equivalent to $(k, R, C)$-DataSAIL. To see why (16) implies (19), note that the right-hand side of (16) is 0 for all $i \in [k]$ if $\pi_\xi(x) = \pi_\xi(x')$. Otherwise, the right-hand side of (16) is 1 for the unique fold $i$ that contains $x$ but not $x'$. Since we minimize over $\boldsymbol{\zeta}$ with non-negative coefficients in the objective, these considerations imply (19).

### Implementation details

We here provide details on the workflow of the heuristic implemented in the DataSAIL Python package and visualized in Fig. 2. In the first step (Fig. 2a), the users can choose between several algorithms to compute similarities or distances for different data types or provide a custom matrix (following Table 2). All distances or similarities are scaled to [0, 1].

Subsequently (Fig. 2b), the input dataset $\mathcal{D}$ is clustered into $K$ clusters, where $K$ is a constant that the user can adjust. This is done separately for the two data types in two-dimensional datasets, leading to $2K$ clusters in total. To cluster similarities, DataSAIL uses spectral clustering[30]; for distances, agglomerative clustering is used[31] (as implemented in scikit-learn[32]).

Using the resulting set of clusters $\mathcal{C}$, DataSAIL then constructs a problem instance of size $K$ (or $2K$ for two-dimensional datasets) as follows (Fig. 2c): The clusters $A \in \mathcal{C}$ act as data elements, inter-cluster similarities or distances $sim_\mathcal{C}, d_\mathcal{C} : \mathcal{C} \times \mathcal{C} \to \mathbb{R}$ are computed using average-, single-, or complete-linkage depending on the choice of the user (if distances are provided, the cluster distances are transformed to cluster similarities as $sim_\mathcal{C} := 1 - dist_\mathcal{C}$). Similarities between entities of different types are currently not supported, i.e., DataSAIL assumes $sim_\mathcal{C}(A, A') = 0$ if $A$ and $A'$ are clusters of data elements of different types. Cardinalities are defined as $\kappa_\mathcal{C}(A) := \sum_{x \in A} \kappa(x)$. The remaining parameters (number of folds $k$, type and class assignments $t$ and $\sigma$, desired

### Table 2 | Algorithms to compute similarities and distances

| Data type | Available algorithms to compute sim or $d$ |
|---|---|
| Proteins | MMseqs2[48], CD-HIT[49,50], FoldSeek[51], user-provided |
| Small molecules | Tanimoto similarity computed for extended-connectivity fingerprints of Bemis-Murcko scaffolds[23–25], user-provided |
| DNA & RNA sequences | CD-HIT[49,50], user-provided |
| Genomes & longer contigs | MASH[52], user-provided |
| Custom data | User-provided |

This table lists algorithms per type of dataset to compute similarities or distances for biochemical data available in DataSAIL.

relative fold sizes $s_i$, error margins $\epsilon$ and $\delta$) are inherited from the input provided by the user. This constant-size instance is then solved by feeding its ILP formulation (Section "An integer linear programming formulation of the $(r, R, C)$-DataSAIL problem") into CVXPY[33-35]—a Python package for convex optimization that provides a unified interface for multiple solvers such as GUROBI[36], MOSEK[37], or SCIP[38].

The ILP solvers return a partition $\pi_{\mathcal{C}} : \mathcal{C} \to [k]$ of the set of clusters $\mathcal{C}$. In the last step (Fig. 2d), this cluster partition is unpacked into a partition $\pi : \mathcal{D} \to [k]$ of the original data points by setting $\pi(x) := \pi_{\mathcal{C}}(A)$ for each $A \in \mathcal{C}$ and each $x \in A$. Note that, by definition of $\kappa_{\mathcal{C}}$, the fact that $\pi_{\mathcal{C}}$ respects the constraints specified in Eqs. (3) and (4) implies that the same constraints are also respected by $\pi$.

### Datasets and machine learning models

For splitting one-dimensional data following $(k, 1, 1)$-DataSAIL, we use the MoleculeNet collection of benchmark datasets with different measured biochemical properties (e.g., toxicity or water solubility), which should be predicted in regression or classification[22]. Since this benchmark contains multiple datasets from different sources, the performance metrics differ between datasets.

An application case for the $(k, 2, 1)$-DataSAIL problem is the LP-PDBBind dataset, comprising experimentally measured binding affinities of 19,443 protein-ligand complexes[39]. To demonstrate the effect of information leakage in stratified splitting, we use the stress response-antioxidant response element (SR-ARE) subchallenge from Tox21[40] as an instance of the $(k, 1, 2)$-DataSAIL problem, where the two classes are active and inactive small molecules in this pathway.

Four classical ML models were trained for all datasets: RF[17], SVM[18], XGB[19], and vanilla multi-layer perceptrons (MLP)[20]. For the one-dimensional datasets, we additionally trained a directed message-passing graph neural network (D-MPNN)[21]. For the two-dimensional dataset LP-PDBBind, we additionally trained DeepDTA, a deep learning model comprising CNN-encoders for proteins and ligands and an MLP predictor based on the encoder outputs[41]. Both deep learning models were selected because they showed top performance in their respective tasks[42,43], do not rely on pre-training (which introduces a new aspect into OOD performance evaluation), and are reasonably easy to use. All models' exact training setups and parameterizations are described in Section "Training of supervised machine learning models".

### Validation protocol

We empirically investigated how DataSAIL improves estimating the performance of ML models on unseen data in two ML tasks: molecular property prediction and drug-target interaction prediction. Here, we compare DataSAIL's similarity-based splitting to random splitting (identity-based splits fulfilling Eq. (3)), fingerprint-based splitting, and LoHi. An extensive comparison against other methods for leakage-reduced data splitting mentioned above is provided in Supplementary Section S1. Details on the hyper-parameters of DataSAIL used for the experiments are given in Section "Hyper-parameter choices".

Using the compared data splitting approaches, we split the benchmark datasets into 80% training and 20% test data (i.e., we set $k = 2$, $s_1 = 0.8$, and $s_2 = 0.2$ for our experiments). We then trained five ML models on the training sets and evaluated their performances on the test sets. We did not need a validation set because we did not tune hyper-parameters. All test performances were averaged over five splittings of the datasets, with shuffling of the dataset between splittings to increase variability. Whenever random data splitting yields consistently better test performances than splitting with DataSAIL, there is evidence for similarity-induced information leakage that can be avoided by similarity-based splitting as implemented in DataSAIL. To better visualize the reduction of information leakage by DataSAIL, we show the average $L(\pi)$ of each splitting algorithm on the right of the performances. For better interpretability and comparability, we define scaled $L(\pi)$ by scaling $L(\pi)$ as defined in Eq. (2) to the interval [0, 1] as

follows:

$$\text{scaled } L(\pi) : = \frac{L(\pi)}{\sum_{x, x' \in \mathcal{D}} sim(x, x')}. \tag{20}$$

### Training of supervised machine learning models

We trained six different models, four of them (RFs, SVMs, XGB, and MLPs) based on the implementation in the `scikit-learn v1.3.2` package. The SVMs and XGB were wrapped in the MultiOutput framework to make them applicable to multi-target learning in a one-versus-all fashion. Following Deng et al.[1], we train the random forest as an ensemble of 500 trees, the SVMs with linear kernels, and XGB with default parameters. For the MLPs, we use 3 hidden layers with sizes 512, 256, and 64 and train them for 200 epochs. Otherwise, all models are trained with default parameters and for reproducibility with `random_state = 42`. The training for these four models was conducted on standard CPUs. The input to all four models for molecular property prediction is a Morgan (ECFP4) fingerprint with a radius 2 hashed to a bit vector size of 1024. When training them on LP-PDBBind splits, we concatenate the Morgan fingerprint of the drug, with radius 2, hashed to a bit vector size of 480, with the ESM-2 embedding of the target. We use the 12-layer ESM-2 model, producing a 480-dimensional protein embedding received from the `fair-esm v2.0.0` Python package.

The fifth model, D-MPNN, was taken from `ChemProp v1.6.1`[44]. As this is a graph neural network, the input is the SMILES string of a molecule. This model was trained with default parameters for 50 epochs on an NVIDIA RTX 3090 with 24 GB GPU RAM. The sixth model is DeepDTA; we used the implementation from the LP-PDBBind GitHub repository (https://github.com/THGLab/LP-PDBBind/). DeepDTA is a state-of-the-art model for drug-target interaction prediction based on two CNN encoders for SMILES and amino acid sequence input. It was trained for 50 epochs with kernel size 8 in both encoders on the same NVIDIA RTX 3090 GPU.

We used three solvers, GUROBI `v11.0.0`, MOSEK `v10.1.21`, and SCIP `v7.0.3`, retrieved through conda. For GUROBI and MOSEK, we issued academic licenses from their respective platforms.

### Hyper-parameter choices

Table 3 summarizes the hyper-parameters and configurations of DataSAIL used to obtain the results reported in this paper. Except for the results reported in Fig. 6a, b, where varying ILP solvers were used

**Table 3 | Hyper-parameter values**

| Figure | Config. | $K$ | $\epsilon$ | $\delta$ | Choice of sim or $d$ |
|---|---|---|---|---|---|
| Figure 3 | S1 | 50 | 0.1 | – | Tanimoto |
| Figure 4 | S1 and S2 | 50 | 0.1 | – | Tanimoto & MMseqs2 |
| Figure 5 | S1 | 50 | 0.1 | 0.1 | Tanimoto |
| Figure 6a, b | S1 | var. | 0.1 | – | Tanimoto |
| Figure 6c | S1 | 50 | var. | var. | Tanimoto |
| Figure 6d | S1 | 50 | 0.1 | – | Tanimoto |
| Supplementary Fig. 1 | S1 | 50 | 0.1 | – | Tanimoto |
| Supplementary Fig. 2 | S1 | 50 | 0.1 | – | Tanimoto |
| Supplementary Fig. 3 | S1 and S2 | 50 | 0.1 | – | Tanimoto & MMseqs2 |
| Supplementary Fig. 4 | S1 and S2 | 50 | 0.1 | – | Tanimoto & MMseqs2 |
| Supplementary Table 1 | S1 | 50 | 0.1 | – | Tanimoto |
| Supplementary Table 2 | S1 and S2 | 50 | 0.1 | – | Tanimoto & FoldSeek |

Details on hyperparameters choices of DataSAIL for the results reported in this work.

and a time limit of 2 h was imposed, all splits were computed with GUROBI and a time limit of 1000 s.

## Data availability

The data from MoleculeNet[22] was fetched through the Python package `deepchem v2.7.1`. Links to download the individual datasets are available at https://moleculenet.org/datasets-1. The data from LP-PDBBind[39] was taken from their GitHub repository (https://github.com/THGLab/LP-PDBBind/). The data for PLINDER[14] was downloaded from the Google Cloud Storage (https://console.cloud.google.com/storage/browser/plinder). The data was taken from v2, and the files determined the splits from v0. It is important to mention that despite the variation in the versions, the benchmark is backward compatible, i.e., data is extended but not altered. Therefore, the v0 splits can be extracted from the v2 data.

## Code availability

All code for DataSAIL and the experiments is available on GitHub at https://github.com/kalininalab/DataSAIL. The code for the experiments is provided in the `experiments` subfolder. Furthermore is the code deposited at Zenodo[45].

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

## Acknowledgements
R.J. and O.V.K. thank Ilya Senatorov, Alexander Gress, and Anne Tolkmitt for fruitful discussions and the members of the Kalinina lab for testing the package. R.J. thanks Daniel Bojar for the opportunity to continue working on DataSAIL during his stay at the BojarLab, University of Gothenburg. R.J. was supported by the HelmholtzAI project XAI-Graph, the Knut and Alice Wallenberg Foundation, and the University of Gothenburg. D.B.B. was funded by the Deutsche Forschungsgemeinschaft (DFG, German Research Foundation, grant no. 516188180), by the German Federal Ministry of Education and Research (BMBF, grant no. 031L0309A and 01KD2419A), and by the Klaus Tschira Foundation (grant no. 00.003.2024). O.V.K. acknowledges financial support from the Klaus Faber Foundation.

## Author contributions
O.V.K. conceived the project. R.J. implemented the Python framework and carried out all experiments. D.B.B. conceived the theory and proved the NP-hardness. D.B.B. and O.V.K. jointly supervised the work. All authors contributed equally to writing and reviewing the manuscript.

## Funding

## Competing interests
D.B.B. consults for BioVariance. The other authors declare no competing interests.
