## [Transparent Peer Review file · Nature Communications]

Data splitting to avoid information leakage with DataSAIL

Corresponding Author: Mr Roman Joeres

Version 0:

Reviewer comments:

Reviewer #1

(Remarks to the Author)

In this work, Joeres et al. present DataSAIL, a method to reduce leakage across data splits. Notably, their implementation of DataSAIL works with 1D and 2D splits. The authors use biochemical data to highlight the effectiveness of DataSAIL relative to existing methods. Overall, this work is important and interesting. I have several suggestions about the framing of the manuscript:

Major:

- 1) The discussion highlights technical advances and only briefly mentions the importance to the field. I believe that, in a journal with a wide audience like Nat Comms, there should be one to several paragraphs describing the potential importance and implications.
- 2) I think it is interesting that the authors frame the problem as data leakage, whereas most previous studies framed it as better evaluating out-of-distribution accuracy. Can the authors justify this decision? Also, I think it would be helpful to briefly highlight this distinction in the introduction or methods.
- 3) It would be helpful if the abstract mentions that DataSAIL is only applicable for models that will be applied to data substantially different from the training data.

Minor:

- 4) Can you make Figure 1 colorblind-friendly please? The red and green colors are necessary for interpretation, but may be difficult for people with colorblindness.
- 5) (Page 8) "Notably, S1 splitting leads to lower RMSE on the test data of all models compared to I1 splittings." Should this say "higher RMSE," as shown in Figures 4C, 4F?

(Remarks on code availability)

The link to "mamba" on the GitHub page does not work. Otherwise, I installed the package and reviewed their documentation.

Reviewer #2

(Remarks to the Author)

The paper provides a approach for splitting data to minimize data leakage. The paper has good analysis. What I am not clear of is the novelty, hence I have the following question

Suggestion / Questions

1. What is $\kappa(x)$ in equation 2 in page 5? $\kappa(x) \in \mathbb{N}_{\geq 1}$ denotes the cardinality of a data element, I am confused. x is a single item, is k is the cardinality or the split size or something else?
2. $s_i \in (0, 1)$, in para above eq 3, if s_i is a value that falls between 0 and 1, it reads like s_i has a value or 0 or 1. Furthermore, cardinality is usually used to indicate size, here is it used to indicate fraction. I would suggest simplifying the terms and math notations to make the paper more readable. Essentially, the argument you are making is to minimize inter-split similarity while keeping the y class proportion similar across the split. My suggestion is to simply the notations and make it readable, seems unnecessarily complex.
3. I am also not sure of what the main novelty or contribution is. How is your approach different to LoHI baseline. Presumably the problem formulation is the same regardless of the method. Or is your novelty the problem formulation?

(Remarks on code availability)

Reviewer #3

(Remarks to the Author)

The paper introduces DataSAIL — a computational tool for data splitting to improve the construction of test sets that better match out-of-distribution samples, and could help researchers get a better understanding of how well an ML model might work in the presence of a real-world distribution shift that matches the type of shift captured by DataSAIL.

My area of expertise is ML methods research. I am not an expert in biomedicine. I am writing my review from this perspective. Data leakage is an important problem in scientific research that uses machine learning, and the paper satisfactorily motivates the focus on this problem.

However, as the authors note in their paper, the type of data leakage addressed by DataSAIL is one of many potential data leakage issues in ML-based science. In particular, it only addresses data leakage when the number of features is small, and the only type of leakage it addresses is papers with test sets that don't match the distribution of the training set.

Upon reading the paper in its current form, I was not convinced that this type of data leakage is a major issue in the field. One of the things that is a must is to justify that this is indeed a prevalent type of data leakage in biomedical sciences, and that this type of leakage leads to a big impact on the reported results. While the paper cites a number of other research studies showing the prevalence of data leakage, as far as I can tell, there is no evidence that the specific type of leakage addressed by DataSAIL is actually an issue in biomedicine.

While I understand that the paper focuses on demonstrating how DataSAIL can improve the evaluation of machine learning models on various datasets, rather than critiquing specific published works, as a researcher from a different field, the authors need to provide specific examples of published works that are affected by the exact type of similarity-induced leakage that DataSAIL addresses to convince me of the utility of the method in real-world applications.

Minor:

- Figure 3 labels are incorrect (A,B are repeated twice)

(Remarks on code availability)

Reviewer #4

(Remarks to the Author)

ML for Drug Discovery is huge and growing, so even the latest Nobel prize was awarded for AlphaFold. Despite the abundance of models and papers, proper benchmarking is lacking. When I read another modeling paper, I have to read between the lines and feel the vibe, because most in-silico benchmarks are hard to interpret. They're full of leaks, their tests don't represent real-life challenges, and I believe they can be gamed. This field has essentially become a competition in overfitting.

This paper correctly identifies the problem, suggests a solution, and compares it with alternatives. It establishes a KPI to measure progress in model development, without which it's extremely hard to know if our models are actually better than those from twenty years ago. That's why I value this kind of methodological paper more than forty new models, and it was an honor to review this important work.

The authors define their goal as minimizing information leakage with respect to stratification constraints, formulate it as an ILP problem, and solve it to split a dataset into training and test sets. To boost performance, they cluster the data points. The authors also study the parameters and compare performance with other methods. The authors explained their methods very clearly, except for some questions I raise below.

The topic is tough. Otherwise, we wouldn't have the problem of 99% of benchmarks having major leakages. Because of that, I think this paper shouldn't just target a methodological audience but also consider readers who routinely use random splits and don't see an issue with it. I'd like to see this paper made more accessible to that audience.

I read the paper and the supplements very thoroughly, except for the 4.1 NP-proof, which I skimmed. It looks fine to me, but I'm not an expert and could've missed something. Still, I think it's not the main contribution, and I'm inclined to believe that a non-trivial ILP formulation is NP-hard indeed.

I'll start with overall comments and suggestions, and then finish with a list of things I consider errors and typos.

DISCUSSION

1. **What is the dataset?** When you introduced dataset D in 2.2, I thought it consisted of tuples like {small molecule, protein, activity} in the R2 situation. After you said, “with a slight abuse of notation, we henceforth let D denote a set of feature vectors instead of a set of feature vector-label pairs,” I thought D was tuples {small molecule, protein}. Then you added cardinality, and I thought D consisted of clusters of the tuples. Much later, I understood it consists of both small molecules and proteins. Is this correct? The hardest part for me was understanding that a dataset point can be either a protein or a small molecule. I suggest making this clearer from the start. For example, you could define D as a collection of all small molecules and proteins in the R2 case.

2. **How to convert the solution of ILP to a split?** If my understanding is correct, the solution provides a mapping from small molecules and proteins to blocks. However, an R2 dataset consists of pairs of small molecules and proteins. How do I map those? What if I have an activity datapoint where a small molecule is assigned to train, but the protein is assigned to test? I found this the most confusing part, and I didn't find the answer in the package documentation. Please clarify.

3. **Simplify Introduction** The reason why we want fair splits is not just because “information leakage is an emerging problem in many fields,” but because we want molecular models that work in real life, not just in in-silico benchmarks. We're mostly on the same page regarding minimizing $L(\pi)$, but I encourage you to make the intro more accessible to someone who has never thought about this problem. Since it's a big issue, it's safe to assume that 90% of readers have never thought beyond a random split and never considered it problematic. I suggest adding more motivation with that kind of reader in mind. This is more of a stylistic suggestion than a technical one.

4. **What is the optimal amount of $L(\pi)$?** We want generalizable models, and minimizing $L(\pi)$ is a good heuristic, which I'm also in favor of. However, we can imagine situations where that heuristic wouldn't be optimal. What if you want to find a GPCR binder and have a ChEMBL dataset? You might find that putting all GPCR binders in the test set and all the rest in the training set decreases $L(\pi)$. In practice, you'd still likely prefer having some GPCR binders in the train set, even if that increased $L(\pi)$, because otherwise you'd get overly pessimistic in-silico metrics.

So, what is the optimal $L(\pi)$ for my specific problem? What's the best $L(\pi)$ for model selection? What's the best $L(\pi)$ to estimate experimental results?

Again, we're mostly aligned, but I'm interested in your thoughts. Keep in mind that your paper will be read by a practitioner who just wants to train models. You mention in the Discussion that minimizing $L(\pi)$ not always appropriate, but could you elaborate more on when it is?

5. **Consistency in performance measures** This relates closely to #4. In the main text, you compare methods by $L(\pi)$, but in Supplement 3, you seem to compare them by model RMSE. You say, “While some tools succeed in reducing information leakage slightly more than S1 splits computed by DataSAIL, they are all clearly outperformed by DataSAIL's S2 splits.” I see that we can't compare S2 and S1 $L(\pi)$, but when do we switch to comparing RMSE? One could imagine an adversarial approach where models are sequentially fit on different splits, data points with the most error are found, and then those are put into the test set. That would maximize RMSE, but I'm not sure that's a good strategy for a split.

All these questions arise because we're minimizing $L(\pi)$ without discussing the exact problem we're solving. Again, this isn't a technical critique but a suggestion to discuss the real-life problem we're trying to solve. If we want to find a GPCR binder, how can we compare different splits, and what's the optimal split in this case?

6. **Additional references**

I suggest adding references to the MOOD (Real-World Molecular Out-Of-Distribution: Specification and Investigation) paper. It specifically discusses previous questions regarding problem formulation and will help make the presentation more accessible to practitioners. The second reference is the Plinder dataset, which presents a ligand-protein benchmark with a better split. I understand that you're competing with their work and your preprint came out a few months before theirs, but please mention and comment on their split methods. It would be awesome if you could compare their $L(\pi)$ (or whatever split quality measure) with DataSAIL's split, but I understand this may be technically difficult, so I don't insist.

7. **Fig 6D** It seems wrong. How can the fingerprint split be 30x faster than the random split? Also, you mention in the supplement regarding fingerprint split “[...] not well scalable to larger datasets, as demonstrated in Figure 6D,” but it's the fastest split on the plot?

8. **Scaled $L(\pi)$**

In 4 Effect of solvers and hyper-parameters, scalability, you mention scaled $L(\pi)$ but don't define it. Please provide a definition. I suspect it's related to #4 and #5 regarding the correct split quality measure.

9. **Tox21 Classes**

In Fig. 6C, you found no effect of delta on $L(\pi)$. The delta is used to stratify the split by classes. What were the classes of the dataset you used?

TYPOS

These are small things that I think are errors. If not, please clarify.

****Fig 1:**** Would be great if you change the font so the “l” doesn't look like an “1”.

****Fig 4, caption, last sentence:**** “The bottom part is ML model performances and information leakages for the different splits a(C, F, and l), measured via the root mean squared error (RMSE, lower is better).”

There's no information leakage on the plots. It should be something like:

“The bottom part shows ML model performances for the different splits (C, F, and l), measured via the root mean squared error (RMSE, lower is better).”

****Page 8, bottom, results of two-dimensional datasets:****

Sentence:

“Notably, S1 splitting leads to lower RMSE on the test data of all models compared to l1 splittings”

It's the other way around: l1 leads to lower RMSE.

****Supplements, DeepChem – Fingerprint****

“The runtime is quadratic and, therefore, not well scalable to larger datasets, as demonstrated in Figure 6D”, but DataSAIL is quadratic as well :)

(Remarks on code availability)

I installed the package and skimmed the docs.

Version 1:

Reviewer comments:

Reviewer #1

(Remarks to the Author)

The authors have improved the manuscript and addressed my concerns. I believe the manuscript should be published.

(Remarks on code availability)

I reviewed the code in the previous iteration of review

Reviewer #2

(Remarks to the Author)

My previous comments have been addressed. I have no further comments

(Remarks on code availability)

Reviewer #3

(Remarks to the Author)

Thank you for the updates. On reading the authors' responses to my and the other reviewers' feedback, I am satisfied that the feedback has been addressed satisfactorily.

(Remarks on code availability)

Reviewer #4

(Remarks to the Author)

Thank you! I love the new introduction, and I'm glad you calculated L(pi) for PLINDER. I confirm that the updates in the 2. Results section have made it clearer with the improved notation.

I noticed one more typo in the supplements (S1.2) regarding the LoHi splitter: "The idea is to remove as few edges as possible [...]" should be "The idea is to remove as few nodes as possible [...]" The new blue text below appears correct.

I recommend this paper for publication and hope this method will contribute to establishing fairer benchmarks in the field.

(Remarks on code availability)

Reviewer #1 (Remarks to the Author):

In this work, Joeres et al. present DataSAIL, a method to reduce leakage across data splits. Notably, their implementation of DataSAIL works with 1D and 2D splits. The authors use biochemical data to highlight the effectiveness of DataSAIL relative to existing methods. Overall, this work is important and interesting. I have several suggestions about the framing of the manuscript:

Major:

1) The discussion highlights technical advances and only briefly mentions the importance to the field. I believe that, in a journal with a wide audience like Nat Comms, there should be one to several paragraphs describing the potential importance and implications.

Answer: We would like to thank the reviewer for prompting us to explain the relevance of our work better. We have completely re-written the Introduction and now highlight more explicitly how avoiding similarity-induced information leakage helps to obtain reliable ML model performance estimates that realistically represent model performance upon deployment. Moreover, in the Discussion, we now more explicitly discuss the relationship between the amount of leaked information, similarity function, and their effect on a model's performance, as well as the limitations of our approach.

2) I think it is interesting that the authors frame the problem as data leakage, whereas most previous studies framed it as better evaluating out-of-distribution accuracy. Can the authors justify this decision? Also, I think it would be helpful to briefly highlight this distinction in the introduction or methods.

Answer: The objectives to better evaluate out-of-distribution (OOD) accuracy and to avoid similarity-induced information/data leakage are closely related, but there is a subtle difference: Similarity-induced information leakage can occur when an ML model that is intended to be used on OOD data during inference is only evaluated on in-distribution data. For ML models that are intended to be used on in-distribution data only, a bad OOD performance is unproblematic. When aiming to develop models that can yield reliable predictions for OOD data in real-world application scenarios, being able to better evaluate OOD accuracy is hence important because it allows to avoid similarity-induced information leakage. In other words: Better evaluating OOD accuracy is a means to avoid information leakage, so we decided to frame the problem in terms of information leakage. We now better explain this in the Introduction:

“In this work, we address the problem of information leakage due to misleading evaluation on in-distribution data for models intended for deployment on OOD data. To this end, we developed DataSAIL, an algorithmic framework and tool to split datasets into multiple folds that allow to realistically estimate model performance on OOD data.”

3) It would be helpful if the abstract mentions that DataSAIL is only applicable for models that will be applied to data substantially different from the training data.

Answer: We added the following clarification to the abstract:

“We present DataSAIL, a versatile Python package to facilitate leakage-reduced data splitting to enable realistic evaluation of ML models for biological data that are intended to be applied in out-of-distribution scenarios.”

Moreover, the last paragraph in the Introduction now starts with the following sentences:

“In this work, we address the problem of information leakage through misleading evaluation of models intended for deployment in OOD scenarios on in-distribution data. To do so, we developed DataSAIL, an algorithmic framework and Python tool to split datasets into training, validation, and test folds that allow to realistically estimate model performance on OOD data.”

Minor:

4) Can you make Figure 1 colorblind-friendly please? The red and green colors are necessary for interpretation, but may be difficult for people with colorblindness.

Answer: Done, thank you for the suggestion.

5) (Page 8) “Notably, S1 splitting leads to lower RMSE on the test data of all models compared to I1 splittings.” Should this say “higher RMSE,” as shown in Figures 4C, 4F?

Answer: Corrected, thank you.

Reviewer #1 (Remarks on code availability):

The link to "mamba" on the GitHub page does not work. Otherwise, I installed the package and reviewed their documentation.

Answer: We updated the link; thanks for mentioning it.

Reviewer #2 (Remarks to the Author):

The paper provides an approach for splitting data to minimize data leakage. The paper has good analysis. What I am not clear of is the novelty, hence I have the following question

Suggestion / Questions

1. What is $\kappa(x)$ in equation 2 in page 5? $\kappa(x) \in \mathbb{N}_{\geq 1}$ denotes the cardinality of a data element, I am confused. x is a single item, is κ the cardinality or the split size or something else?

Answer: We have substantially re-written Section 2.2 to better introduce the mathematical notation. We now write:

“More formally, let \mathcal{D} be the set of data points represented in \mathcal{M} or a set of clusters defined over these data points (allowing clusters as elements of \mathcal{D} will be important for our heuristic solver, as explained below). [...] Moreover, the data points or clusters $x \in \mathcal{D}$ have cardinalities $\kappa(x) \in \mathbb{N}_{\geq 1}$. For elementary data points, we always have $\kappa(x) = 1$; if \mathcal{D} contains clusters, we may have $\kappa(x) > 1$.”

We now clearly differentiate between the set of tuples \mathcal{M} and the set of corresponding data points/clusters \mathcal{D} . The cardinality refers, in this case, to the data points/clusters, not tuples or splits. We hope that this clarifies this question.

2. $s_i \in (0, 1)$, in para above eq 3, if s_i is a value that falls between 0 and 1, it reads like s_i has a value or 0 or 1. Furthermore, cardinality is usually used to indicate size, here is it used to indicate fraction. I would suggest simplifying the terms and math notations to make the paper more readable.

Answer: The notation “ $s_i \in (0, 1)$ ” here refers to s_i being in the open interval $(0, 1)$. This is standard mathematical notation; if s_i was constrained to be either 0 or 1, we would have written “ $s_i \in \{0, 1\}$ ”. In the original version of the manuscript, we had referred to s_i as “relative cardinality”. We acknowledge that this may be confusing and therefore now call s_i “split fraction”. Moreover, we have added the following explanation after Eq. 3:

“For elementary data points where $\kappa(x) = 1$, this means that the fraction of data points of type r (that is, the data points in $\mathcal{D}_{t=r}$) that are assigned to split i (the data points in \mathcal{D}_i^π) matches the desired relative split sizes up to a relative error ϵ .”

Essentially, the argument you are making is to minimize inter-split similarity while keeping the y class proportion similar across the split. My suggestion is to simply the notations and make it readable, seems unnecessarily complex.

Answer: We need to introduce the precise mathematical notation for the proof that underlies our NP-hardness result. To improve readability, we have re-written large parts of Section 2.2, where we introduce the (k, R, C) -DataSAIL problem. In particular, we have added the following high-level summary at the beginning of Section 2.2, relying on the reviewer’s suggested summary of our idea:

“Intuitively, we define (k, R, C) -DataSAIL as the problem to minimize inter-class similarity while keeping similar class ratios across the splits.”

3. I am also not sure of what the main novelty or contribution is. How is your approach different to LoHi baseline. Presumably the problem formulation is the same regardless of the method. Or is your novelty the problem formulation?

Answer: There are two main novelties w.r.t. the Lo-Hi baseline:

- We suggest a new formulation of the data splitting problem which allows us to compute stratified splits for multi-dimensional data. In contrast, the Balanced Vertex Minimum k -Cut Problem underlying Lo-Hi only supports non-stratified splits on one-dimensional data.
- DataSAIL is substantially more flexible than Lo-Hi in that it supports various types of biological data, whereas Lo-Hi can only be used for small molecule data.

We summarize DataSAIL’s novelty w.r.t. existing approaches at the end of the substantially revised Introduction, where we write:

“Unlike existing tools and algorithms to compute data splits that reduce information leakage, DataSAIL can automatically compute splits for heterogeneous data of two different types and combines stratification with similarity-aware splitting (Table 1, see supplement for comparison with existing tools). Moreover, DataSAIL is more versatile than existing approaches because it can be used out-of-the-box for various types of molecular data.”

Table 1 provides an overview of how DataSAIL differs from Lo-Hi and other data-splitting approaches. For more in-depth comparisons against these existing methods, we have added additional explanations to the section “Detailed description of related work” in the supplement.

Reviewer #3 (Remarks to the Author):

The paper introduces DataSAIL — a computational tool for data splitting to improve the construction of test sets that better match out-of-distribution samples, and could help researchers get a better understanding of how well an ML model might work in the presence of a real-world distribution shift that matches the type of shift captured by DataSAIL.

My area of expertise is ML methods research. I am not an expert in biomedicine. I am writing my review from this perspective. Data leakage is an important problem in scientific research that uses machine learning, and the paper satisfactorily motivates the focus on this problem.

However, as the authors note in their paper, the type of data leakage addressed by DataSAIL is one of many potential data leakage issues in ML-based science. In particular, it only addresses data leakage when the number of features is small, and the only type of leakage it addresses is papers with test sets that don't match the distribution of the training set.

Answer: DataSAIL indeed only addresses data leakage that occurs when the distribution of the data seen at inference time does not match the distribution of the training data. We now clarify this in the Introduction as follows:

“In this work, we address the problem of information leakage through misleading evaluation of models intended for deployment in OOD scenarios on in-distribution data. To do so, we developed DataSAIL, an algorithmic framework and Python tool to split datasets into training, validation, and test folds that allow to realistically estimate model performance on OOD data.”

The number of features is irrelevant to the applicability of DataSAIL or its outcome. Do some of the explanations we provide in our manuscript misleadingly suggest that DataSAIL can only be used when the number of features is small? If so, we'd be grateful if Reviewer 3 could point us to the specific text passages such that we can improve our explanations.

Upon reading the paper in its current form, I was not convinced that this type of data leakage is a major issue in the field. One of the things that is a must is to justify that this is indeed a prevalent type of data leakage in biomedical sciences, and that this type of leakage leads to a big impact on the reported results. While the paper cites a number of other research studies showing the prevalence of data leakage, as far as I can tell, there is no evidence that the specific type of leakage addressed by DataSAIL is actually an issue in biomedicine.

While I understand that the paper focuses on demonstrating how DataSAIL can improve the evaluation of machine learning models on various datasets, rather than critiquing specific published works, as a researcher from a different field, the authors need to provide specific examples of published works that are affected by the exact type of similarity-induced leakage that DataSAIL addresses to convince me of the utility of the method in real-world applications.

Answer: As also pointed out by Reviewer 4, the type of data leakage addressed by DataSAIL is indeed a major problem in biomedical ML research, and we would like to thank Reviewer 3 for prompting us to better explain this to readers from the ML community. In the Introduction, we now provide two specific examples – protein-protein interaction (PPI) prediction and deleteriousness prediction for missense variants – where similarity-induced data leakage as addressed by DataSAIL has led to numerous overly optimistic performance estimates in published works: For the case of deleteriousness prediction, Grimm *et al.*

(<http://dx.doi.org/10.1002/humu.22768>) showed that, when variants which are similar in that they affect the same protein are assigned to different splits, ML models can achieve excellent test performances by relying on protein-level shortcuts (e. g., a simple protein-level majority vote based on the variants in the training fold). Such models then generalize poorly to variants of sparsely annotated proteins and will systematically misclassify minority-class variants of proteins for which both deleterious and non-deleterious variants are seen at inference time. For the case of PPI prediction, Bennett *et al.* (<http://dx.doi.org/10.1093/bib/bbae076>) showed that although published deep learning models perform excellently when evaluated on the random data splits used in the original publications, performances often become close to random when the evaluation is performed in a low-homology setting that represents the use case to predict PPIs for poorly characterized proteins. This is now explained in the Introduction.

Additionally, we now also compare the performance estimates achieved by the D-MPNN model for the prediction of molecular properties on the data splits computed by DataSAIL to the performance estimates reported in the original D-MPNN publication (Yang *et al.* (<http://dx.doi.org/10.1021/acs.jcim.9b00237>)). In the original publication, D-MPNN was evaluated on a random split and a scaffold split. The comparison is shown in the newly added Supplementary Table 1. DataSAIL's splits yield similarly hard generalization problems as the scaffold splits constructed by the authors of D-MPNN, indicating that out-of-the-box data splitting with DataSAIL can realistically represent relevant OOD scenarios.

Minor:

- Figure 3 labels are incorrect (A,B are repeated twice)

Answer: Thank you for pointing this out, we fixed the labels.

Reviewer #4 (Remarks to the Author):

ML for Drug Discovery is huge and growing, so even the latest Nobel prize was awarded for AlphaFold. Despite the abundance of models and papers, proper benchmarking is lacking. When I read another modeling paper, I have to read between the lines and feel the vibe, because most in-silico benchmarks are hard to interpret. They're full of leaks, their tests don't represent real-life challenges, and I believe they can be gamed. This field has essentially become a competition in overfitting.

This paper correctly identifies the problem, suggests a solution, and compares it with alternatives. It establishes a KPI to measure progress in model development, without which it's extremely hard to know if our models are actually better than those from twenty years ago. That's why I value this kind of methodological paper more than forty new models, and it was an honor to review this important work.

The authors define their goal as minimizing information leakage with respect to stratification constraints, formulate it as an ILP problem, and solve it to split a dataset into training and test sets. To boost performance, they cluster the data points. The authors also study the parameters and compare performance with other methods. The authors explained their methods very clearly, except for some questions I raise below.

The topic is tough. Otherwise, we wouldn't have the problem of 99% of benchmarks having major leakages. Because of that, I think this paper shouldn't just target a methodological audience but also consider readers who routinely use random splits and don't see an issue with it. I'd like to see this paper made more accessible to that audience.

I read the paper and the supplements very thoroughly, except for the 4.1 NP-proof, which I skimmed. It looks fine to me, but I'm not an expert and could've missed something. Still, I think it's not the main contribution, and I'm inclined to believe that a non-trivial ILP formulation is NP-hard indeed.

I'll start with overall comments and suggestions, and then finish with a list of things I consider errors and typos.

Answer: We would like to thank Reviewer 4 for the positive assessment of our work and the extremely helpful suggestions on how to further improve it.

DISCUSSION

1. ****What is the dataset?*** When you introduced dataset D in 2.2, I thought it consisted of tuples like {small molecule, protein, activity} in the R2 situation. After you said, "with a slight abuse of notation, we henceforth let D denote a set of feature vectors instead of a set of feature vector-label pairs," I thought D was tuples {small molecule, protein}. Then you added cardinality, and I thought D consisted of clusters of the tuples. Much later, I understood it consists of both small molecules and proteins. Is this correct? The hardest part for me was understanding that a dataset point can be either a protein or a small molecule. I suggest making this clearer from the start. For example, you could define D as a collection of all small molecules and proteins in the R2 case.

Answer: This was indeed confusing. Thanks a lot for prompting us to explain it better (the "slight abuse of notation" wasn't that slight, after all). We now use two separate symbols, \mathcal{M} and \mathcal{D} . $\mathcal{M} = \{(x_1, y_1), \dots, (x_n, y_n)\}$ denotes the set of pairs of feature vectors and response variables. In the one-dimensional case, the feature vectors x_i and the response variables y_i represent one data element; in the two-dimensional case, they both represent two data elements. We now clarify this in Section 2.1 as follows:

"If the feature vector x_i and the output measurement y_i are associated with two elementary data points (such as in drug-target interaction prediction, where x_i represents a pair of a molecule and a protein target for which an interaction affinity y_i should be predicted), we call this a *two-dimensional* dataset (Figure 1, bottom)."

As suggested by the reviewer, in Section 2.2, we now explicitly define \mathcal{D} as the set of data points represented by the feature vectors in \mathcal{M} or a set of clusters defined over these data points. We write:

"More formally, let \mathcal{D} be the set of data points represented in \mathcal{M} or a set of clusters defined over these data points (allowing clusters as elements of \mathcal{D} will be important for our heuristic solver, as explained below). The data points/clusters $x \in \mathcal{D}$ can have $R \in \mathbb{N}$ different entity types $t(x) \in [R] := \{1, \dots, R\}$. For instance, in a drug-target interaction scenario, we have $R = 2$ with the two entity types $r \in \{1, 2\}$ corresponding to drugs and protein targets or clusters thereof. If \mathcal{M} is a one-dimensional dataset, all data points/clusters have the same element type. Moreover, the data points or clusters $x \in \mathcal{D}$ have cardinalities $\kappa(x) \in \mathbb{N}_{\geq 1}$. For elementary data points, we always have $\kappa(x) = 1$; if \mathcal{D} contains clusters, we may have $\kappa(x) > 1$."

2. ****How to convert the solution of ILP to a split?*** If my understanding is correct, the solution provides a mapping from small molecules and proteins to blocks. However, an R2 dataset consists of pairs of small molecules and proteins. How do I map those? What if I

have an activity datapoint where a small molecule is assigned to train, but the protein is assigned to test? I found this the most confusing part, and I didn't find the answer in the package documentation. Please clarify.

Answer: Thanks a lot for this comment. We indeed didn't explain this properly in the original manuscript (also because we hadn't distinguished between \mathcal{M} and \mathcal{D}). We added the following explanation at the end of Section 2.2:

“Once a mapping π has been computed for \mathcal{D} , we can use it to compute a mapping for \mathcal{M} : First, we unpack π and assign each data element z contained in the cluster $x \in \mathcal{D}$ to $\pi(x)$, i.e., we define $\pi(z) := \pi(x)$ for all $z \in x$ (Figure 2D). Then, we assign the feature vector-label pair $(x_j, y_j) \in \mathcal{M}$ to the split i if and only if all $\pi(z) = i$ holds for all data points z represented by x_j (e.g., a drug and a protein in the case of drug-target interaction prediction). Feature vector-label pairs (x_j, y_j) with conflicting assignments for different data points represented by x_j are discarded. For instance, in a drug target prediction scenario, it may happen that the drug and the protein jointly represented by x_j are assigned to different splits by π (e.g., the drug is assigned to the training fold and the protein is assigned to the test fold). In this case, (x_j, y_j) is discarded.”

For 2D datasets, it may hence happen that some of the data in \mathcal{M} is lost. To make this explicit, we have added a row “Preserves all data (2D splits)” to Table 1 with entry “No” for DataSAIL and entries “N/A” for all other data-splitting tools.

3. ****Simplify Introduction**** The reason why we want fair splits is not just because “information leakage is an emerging problem in many fields,” but because we want molecular models that work in real life, not just in in-silico benchmarks. We're mostly on the same page regarding minimizing $L(\pi)$, but I encourage you to make the intro more accessible to someone who has never thought about this problem. Since it's a big issue, it's safe to assume that 90% of readers have never thought beyond a random split and never considered it problematic. I suggest adding more motivation with that kind of reader in mind. This is more of a stylistic suggestion than a technical one.

Answer: We have completely re-written the Introduction to make it more accessible to wider audiences. For instance, we now describe two specific examples of biomedical ML problems where simple random splitting has been found to lead to similarity-induced data leakage (protein-protein interaction prediction and deleteriousness prediction for missense variants, see reply to second comment of Reviewer 3 for details). We now also write at the very beginning of the Introduction that fair splits are crucial to ensure generalizability to data seen in real-world deployment scenarios:

“For successful deployment of these ML models in real-world use cases, it is crucial that the reported performance estimates reliably represent model performance during inference. If the test set used for model evaluation does not represent the data used at inference time, the model can achieve inflated performance scores, impeding successful model deployment in a real-world use case. Such misrepresentation can happen when the model uses information from the training set at test time, although this information is not available during inference. This phenomenon is called information leakage or data leakage.”

4. ****What is the optimal amount of $L(\pi)$?**** We want generalizable models, and minimizing $L(\pi)$ is a good heuristic, which I'm also in favor of. However, we can imagine situations where that heuristic wouldn't be optimal. What if you want to find a GPCR binder and have a ChEMBL dataset? You might find that putting all GPCR binders in the test set and all the rest in the training set decreases $L(\pi)$. In practice, you'd still likely prefer having some GPCR binders in the train set, even if that increased $L(\pi)$, because otherwise you'd get overly pessimistic in-silico metrics.

So, what is the optimal $L(\pi)$ for my specific problem? What's the best $L(\pi)$ for model selection? What's the best $L(\pi)$ to estimate experimental results?

Again, we're mostly aligned, but I'm interested in your thoughts. Keep in mind that your paper will be read by a practitioner who just wants to train models. You mention in the Discussion that minimizing $L(\pi)$ not always appropriate, but could you elaborate more on when it is?

Answer: Thank you for these very good questions that made us consider some concrete examples and suggestions for the Discussion section. We added the following considerations at the end of the Discussion, which will hopefully help practitioners to correctly apply DataSAIL:

“The leakage function $L(\pi)$ minimized by DataSAIL becomes small when $\text{sim}(x, x')$ is small for data elements x and x' that π assigns to different splits. Since DataSAIL allows the user to select from various pre-implemented similarity functions sim and is open to custom similarity functions, the user has full control over the behavior of $L(\pi)$. Which choice of sim is most appropriate depends on the intended deployment scenario for the evaluated ML model. In particular, if the inference-time data is expected to be similar to the training data with respect to the similarity function sim selected by the user, evaluating an ML model on the data splits computed by DataSAIL will lead to overly pessimistic results. When using DataSAIL to compute splits for evaluating an ML model that is intended to generalize to OOD data, model evaluators hence have to ensure that the selected similarity function sim indeed captures the intended generalization task.”

Moreover, Reviewer 4 correctly remarked that it may happen that the selected similarity function sim is correlated with the response variable to be predicted by the ML model (e.g., GPCR binders may be more similar to other GPCR binders than to non-binders). In such scenarios, it is crucial that the user runs DataSAIL with the stratification constraint (Equation 4 in Section 2.2), with the classes c defined in terms of the response variable. Without such a constraint, DataSAIL would compute splits that are highly imbalanced w.r.t. the response variable, which may again lead to overly pessimistic performance estimates.

5. ****Consistency in performance measures**** This relates closely to #4. In the main text, you compare methods by $L(\pi)$, but in Supplement 3, you seem to compare them by model RMSE. You say, “While some tools succeed in reducing information leakage slightly more than S1 splits computed by DataSAIL, they are all clearly outperformed by DataSAIL's S2 splits.” I see that we can't compare S2 and S1 $L(\pi)$, but when do we switch to comparing RMSE? One could imagine an adversarial approach where models are sequentially fit on different splits, data points with the most error are found, and then those are put into the test set. That would maximize RMSE, but I'm not sure that's a good strategy for a split.

All these questions arise because we're minimizing $L(\pi)$ without discussing the exact problem we're solving. Again, this isn't a technical critique but a suggestion to discuss the real-life problem we're trying to solve. If we want to find a GPCR binder, how can we compare different splits, and what's the optimal split in this case?

Answer: We agree that the relationship between $L(\pi)$ and the models' performance should be discussed more explicitly and thus added the following passage to the Discussion:

“Given an appropriate choice of sim , a positive correlation between $L(\pi)$ and performance then indicates that the tested ML models struggle to generalize the OOD scenarios modeled by sim .”

Moreover, we have rewritten large parts of the Results section and sections of the supplement where we describe the additional results. The new presentation of the results now makes it more explicit (i) that $L(\pi)$ is the primary performance measure and (ii) that, independently of the employed data splitting algorithm, low values of $L(\pi)$ tend to go hand-in-hand with drops in model performance.

6. ****Additional references****

I suggest adding references to the MOOD (Real-World Molecular Out-Of-Distribution: Specification and Investigation) paper. It specifically discusses previous questions regarding problem formulation and will help make the presentation more accessible to practitioners. The second reference is the Plinder dataset, which presents a ligand-protein benchmark with a better split. I understand that you're competing with their work and your preprint came out a few months before theirs, but please mention and comment on their split methods. It would be awesome if you could compare their $L(\pi)$ (or whatever split quality measure) with DataSAIL's split, but I understand this may be technically difficult, so I don't insist.

Answer: Thank you for bringing MOOD to our attention. We added it as an additional reference in the Introduction.

Following your suggestion, we applied DataSAIL to the PLINDER database to split the data from version 0, which was used for the preprint. We split the PLINDER_NR dataset using DataSAIL's S1 and S2 splits and compare the leakage of these splits to the leakage of the PLINDER-PL50, PLINDER-ECOD, and PLINDER-TIME splits. We included the results below the LP-PDBBind results in the main text and at the end of

7. ****Fig 6D**** It seems wrong. How can the fingerprint split be 30x faster than the random split? Also, you mention in the supplement regarding fingerprint split “[...] not well scalable to larger datasets, as demonstrated in Figure 6D,” but it's the fastest split on the plot?

Answer: Thank you for pointing this out, we re-evaluated Fig. 6D and updated it, removing the wrong claims about the runtime. Still, the random split is indeed slower than the scaffold-based and weight-based splits from DeepChem. We computed I1 splits with DataSAIL (equivalent to a random split on one-dimensional datasets). This I1 split is slower than the two splits from DeepChem, because DataSAIL runs a duplicate detection and solves an ILP considering potential sample weights.

8. ****Scaled $L(\pi)$ ****

In 4 Effect of solvers and hyper-parameters, scalability, you mention scaled $L(\pi)$ but don't define it. Please provide a definition. I suspect it's related to #4 and #5 regarding the correct split quality measure.

Answer: We define it at the bottom of Section 4.4. The scaling is done by dividing $L(\pi)$ by the total leakage in the unsplit dataset. We added references to this section where mentioned. We also updated all figures to explicitly state they show Scaled $L(\pi)$.

9. **Tox21 Classes**

In Fig. 6C, you found no effect of delta on $L(\pi)$. The delta is used to stratify the split by classes. What were the classes of the dataset you used?

Answer: The Tox21 dataset is a dataset with multiple binary targets. We used the SR-ARE target and defined the classes as the labels of this binary target. In the stratification, we balanced positive and negative samples in both splits. We added a better explanation for this in the legend of Figure 5.

TYPOS

These are small things that I think are errors. If not, please clarify.

Fig 1: Would be great if you change the font so the “l” doesn't look like an “1”.

Answer: Good suggestion, we updated the figure accordingly. Thank you.

Fig 4, caption, last sentence: “The bottom part is ML model performances and information leakages for the different splits (C, F, and I), measured via the root mean squared error (RMSE, lower is better).”

There's no information leakage on the plots. It should be something like:

“The bottom part shows ML model performances for the different splits (C, F, and I), measured via the root mean squared error (RMSE, lower is better).”

Answer: We updated the plot, and now there is information leakage in the plot. Thanks for mentioning this.

Page 8, bottom, results of two-dimensional datasets:

Sentence:

“Notably, S1 splitting leads to lower RMSE on the test data of all models compared to I1 splittings”

It's the other way around: I1 leads to lower RMSE.

Answer: Yes, we corrected this. Thank you.

Supplements, DeepChem – Fingerprint

“The runtime is quadratic and, therefore, not well scalable to larger datasets, as demonstrated in Figure 6D”, but DataSAIL is quadratic as well :)

Answer: Yes, you are right. Thank you for pointing it out. We removed all claims of runtime superiority of DataSAIL from the manuscript.

Reviewer #4 (Remarks on code availability):

I installed the package and skimmed the docs.

Reviewer #1 (Remarks to the Author):

The authors have improved the manuscript and addressed my concerns. I believe the manuscript should be published.

Answer: We thank the reviewer for the comments and support.

Reviewer #1 (Remarks on code availability):

I reviewed the code in the previous iteration of review

Reviewer #2 (Remarks to the Author):

My previous comments have been addressed. I have no further comments

Answer: We thank the reviewer for the comments and support.

Reviewer #3 (Remarks to the Author):

Thank you for the updates. On reading the authors' responses to my and the other reviewers' feedback, I am satisfied that the feedback has been addressed satisfactorily.

Answer: We thank the reviewer for the comments and support.

Reviewer #4 (Remarks to the Author):

Thank you! I love the new introduction, and I'm glad you calculated $L(\pi)$ for PLINDER. I confirm that the updates in the 2. Results section have made it clearer with the improved notation.

I noticed one more typo in the supplements (S1.2) regarding the LoHi splitter: "The idea is to remove as few edges as possible [...]" should be "The idea is to remove as few nodes as possible [...]." The new blue text below appears correct.

Answer: Thank you for pointing this out. This was indeed wrong, and we corrected the text.

I recommend this paper for publication and hope this method will contribute to establishing fairer benchmarks in the field.

Answer: We thank the reviewer for the comments and support.